# Which Structural Interventions for Adolescent Contraceptive Use Have Been Evaluated in Low- and Middle-Income Countries?

**DOI:** 10.3390/ijerph191811715

**Published:** 2022-09-16

**Authors:** Helen Elizabeth Denise Burchett, Dylan Kneale, Sally Griffin, Málica de Melo, Joelma Joaquim Picardo, Rebecca S. French

**Affiliations:** 1Department of Public Health, Environments & Society, Faculty of Public Health & Policy, London School of Hygiene & Tropical Medicine, London WC1E 7HT, UK; 2EPPI-Centre, UCL Social Research Institute, University College London, London WC1H 0NR, UK; 3International Center for Reproductive Health: Mozambique, Maputo, Mozambique

**Keywords:** contraception, family planning, adolescent, structural, upstream, intervention evaluation, cash transfer, schooling, norms, empowerment

## Abstract

Reducing adolescent childbearing is a global priority, and enabling contraceptive use is one means of achieving this. Upstream factors, e.g., gender inequalities, fertility norms, poverty, empowerment and schooling, can be major factors affecting contraceptive use. We conducted a systematic map to understand which structural adolescent contraception interventions targeting these upstream factors have been evaluated in LMICs. We searched eight academic databases plus relevant websites and a 2016 evidence gap map and screened references based on set inclusion criteria. We screened 6993 references and included 40 unique intervention evaluations, reported in 138 papers. Seventeen evaluations were reported only in grey literature. Poverty reduction/economic empowerment interventions were the most common structural intervention, followed by interventions to increase schooling (e.g., through legislation or cash transfers) and those aiming to change social norms. Half of the evaluations were RCTs. There was variation in the timing of endline outcome data collection and the outcome measures used. A range of structural interventions have been evaluated for their effect on adolescent contraceptive use/pregnancy. These interventions, and their evaluations, are heterogenous in numerous ways. Improved understandings of how structural interventions work, as well as addressing evaluation challenges, are needed to facilitate progress in enabling adolescent contraceptive use in LMICs.

## 1. Introduction

Reducing adolescent childbearing is a global priority and an indicator for Sustainable Development Goal 3, “to ensure healthy lives and promote well-being for all at all ages” [1,2]. Contraceptive use is one means of achieving this by enabling people to choose the timing of planned pregnancies, to attain the desired number of children and to allow spacing between pregnancies to improve the health status of women and their children. Whilst barriers to contraceptive use are experienced by all ages, there is evidence that this is more likely to be felt by adolescent girls and young women (hereafter referred to as adolescent girls) than older women [3]. Unmet need for contraception, when a woman who is sexually active, fecund and does not wish to conceive at that time is not currently using any modern method, is typically higher for adolescent girls aged 15–19 years compared to those aged 20–24 years in low- and middle-income countries (LMIC) [3].

To date, the focus of reviews on the effectiveness of interventions to encourage adolescent contraceptive use has typically been on the supply of contraceptives and services, and/or individual-level demand-side factors [4,5,6,7,8,9,10]. Yet we know that upstream factors, such as gender inequalities, fertility norms, poverty, girls’ empowerment and schooling, can also be major factors affecting contraceptive use [11]. Given the strong influence that these factors can have on an individual’s knowledge, attitudes and behaviours, interventions that address these issues have the potential to have a greater impact than those targeting individual-level factors alone. Structural interventions target the structural-level factors, i.e., “the physical, social, cultural, organizational, community, economic, legal, or policy aspects of the environment” (p1) that can affect health and contraceptive behaviours [12]. 

Although the importance of upstream factors has been recognised [13,14], much research has focused on evaluating interventions targeting adolescents’ knowledge, beliefs, attitudes and skills rather than structural interventions that target these wider determinants [15,16]. For example, an evidence gap map of adolescent reproductive and sexual health impact evaluations and systematic reviews by 3ie found that the most frequently evaluated intervention type was sexual health education [17].

As part of an evidence synthesis project funded by CEDIL, we conducted a systematic map to understand what types of structural adolescent contraception interventions have been evaluated in LMICs. 

## 2. Materials and Methods

Rather than duplicate the comprehensive searches and screening conducted for the evidence gap map by 3ie, mentioned above [17], we screened all the impact evaluations they included and then conducted a systematic search from 2016 to July 2020 in eight databases, using controlled and free-text terms relating to adolescence, family planning and LMICs (see Appendix A for full details of the search strategy). Due to language proficiency within the team, searches were limited to English or Portuguese language references. We limited included papers to those published in 2005 or later, since it was then that global interest in contraceptive use grew [18] as well as evaluations of structural sexual and reproductive health interventions [17]. We used the WHO’s definition of adolescence, i.e., 10–19 years [19] and the World Bank’s definition of low- or middle-income country [20]. In addition, grey literature was sought from 16 websites (see Appendix A) and reference lists from relevant systematic reviews were screened. 

Search results were downloaded into Endnote and duplicates were removed before being uploaded into EPPI-Reviewer for screening. Each reference was screened for potential inclusion on the basis of title and abstract, using pre-specified exclusion criteria to ensure relevance (see Table 1). 

An initial sub-set of references were screened by four researchers (H.B., S.G., M.M., J.J.P.) to ensure consistency of understanding and application of criteria. Once at least 80% consistency had been achieved, the remaining references were screened by individual researchers. For those included at the title/abstract screening stage, full reports were obtained and screened by two researchers (H.B. and either S.G., M.M., J.J.P. or D.K.). Where agreement could not be reached, the paper was discussed with a third researcher. 

Where an intervention evaluation had been reported in multiple papers, these were identified as linked and one paper designated the main paper, to avoid duplicate counting.

A standardised coding tool was developed by the team to capture basic information about the study and the intervention, e.g., country, intervention activities, population, study design and outcomes reported. All included studies were coded using this tool. 

## 3. Results

We screened 6993 references on title/abstract and excluded 6727, then retrieved and screened the full text of 250; we were not able to retrieve 16 references (see Appendix B for PRISMA flow diagram). 

In total, 40 intervention evaluations were included, reported in 138 papers (i.e., 98 papers were secondary or subsequent to the main included paper) (see Appendix C for table of characteristics).

The majority of interventions were evaluated in Africa (24 studies), followed by Asia (n = 8) and South America (n = 6) and the Middle East (n = 3); five studies were multi-country. Five studies were conducted in India and in Kenya, four in Malawi and three each in Mexico, Zimbabwe and Uganda.

Seventeen of the forty intervention evaluations were reported only in grey literature.

### 3.1. Aims of the Interventions

Although to be included, studies had to report pregnancy, birth or contraceptive outcomes, only half of the interventions (n = 20) aimed to increase contraceptive use or improve sexual and reproductive health (implicitly or explicitly including contraceptive use). Another eight interventions aimed to prevent HIV infection, delay early marriage or reduce sexual abuse but did not specifically focus on contraceptive use. In just under a third of studies (n = 12), the intervention had other primary aims, such as increasing participation in education, or reducing poverty.

### 3.2. Type of Structural Interventions

A range of structural interventions were evaluated, often combined with non-structural activities, such as health service provider training or mass media campaigns (see Table 2). Most involved activities that implicitly or explicitly aimed to reduce poverty or increase economic empowerment (n = 29) or aimed to encourage participation in school (n = 17). Thirteen interventions aimed to change social norms within the community.

Although we did not consider “safe space” interventions to be structural interventions themselves, half of the structural interventions (n = 20) that we included had a safe space component. Safe space groups were where girls could meet regularly, often with a mentor (typically a slightly older woman from the community), for education, training and/or recreational purposes. We considered interventions to have a safe space component if they either explicitly described themselves as such, or if they were girls-only groups which mentioned that one of their aims was to increase girls’ social/peer networks. Although many safe space interventions followed a similar format, their content as well as their frequency and duration varied, with most meeting weekly, e.g., in the Safe and Smart Savings Products for Vulnerable Adolescent Girls program [21] or several times a week, e.g., the ELA—Tanzania program [22]; in one intervention, the First Time Parents Project, participants met monthly [23]. Activities in these safe space groups could include literacy and numeracy lessons, life skills, sexual and reproductive health education, other health education (e.g., nutrition), vocational training, financial literacy, savings account activities, community development projects, sport and recreation (see Appendix B for details). Other evaluated interventions involved small group activities, but were not considered to include a safe space component as these were not described as creating “safe spaces” for adolescent girls, nor did they explicitly aim to increase their social networks, e.g., Regai Dzive Shiri [24].

Poverty reduction/economic empowerment interventions were the most common type of structural activity in the evaluations. These interventions included different activities: financial literacy training, vocational or livelihood training, the provision of conditional or unconditional cash or non-cash transfers, microfinance, the creation of savings accounts for girls or the provision of employment opportunities (see Table 3). 

Vocational and livelihoods training activities varied from those offering girls insights into potential employment options in order to raise their aspirations, e.g., in BALIKA [25], to six-month-long vocational training courses at local training institutes, followed by a micro-grant for those who completed the training and developed a business plan, as in SHAZ! [26]. 

Ten of the twelve interventions that incorporated financial literacy training delivered this through “safe space” groups, e.g., in the Ishraq program [27]. 

Cash transfers and all of the non-cash transfers were generally provided to the adolescent girls’ household rather than to the girls directly. Most cash transfers were conditional on girls’ enrolment in, or sufficient attendance at, school, e.g., the Punjab Female School Stipend Program [28]. Other cash transfers were conditional on attendance at the intervention sessions, e.g., Girl Power—Malawi [29]. Non-cash transfers included 50 kg of lentils every six months conditional on attendance at 80+% of the intervention’s meetings (Sawki [30]), a goat at the end of a two-year intervention (Berhane Hewan [31]) or cooking oil every four months, conditional on the adolescent girl remaining unmarried (Kishoree Kontha [32]). Although some interventions offered vouchers for health services (e.g., AGI-K, Marriage: No Child’s Play [33,34]), or school supplies (e.g., Berhane Hewan, Zimbabwean comprehensive school support intervention, Kenyan school subsidies and teacher training intervention [31,35,36]), these were not considered non-cash transfers as they had limited financial value. 

Seventeen studies evaluated interventions that aimed to increase schooling, either through legislative changes (e.g., extending compulsory primary school education [37] or removing schools fees, as in the Universal Primary Education Program [38]), conditional cash transfers as in the Punjab Female School Stipend Program [28], payment of school fees [39], provision of school supplies (e.g., uniforms) [35] or working with schools, parents and/or communities to support girls re-joining or remaining i, school, e.g., Marriage: No Child’s Play [33]. 

Thirteen studies explicitly aimed to change community or social norms around gender, fertility or sexual and reproductive health issues, although others may also have aimed to do so implicitly. Activities were mostly some form of community meetings and dialogue, such as “community conversations”, e.g., in Marriage—No Child’s Play [33]. Others involved community groups working through a programme, such as in Regai Dzive Shiri [24], or developing their own action plan, such as in the Ishraq pilot and scale-up [27,40].

Most interventions lasted between 18 months and 3 years, although a few were shorter, e.g., Girl Empower—Malawi [41], or longer, e.g., the Ghanaian School Scholarship Programme [39]; for some, the duration was not clear or varied, particularly those that were government cash transfer schemes, e.g., Oportunidades [42].

### 3.3. Who Was Targeted by the Intervention?

All of the interventions targeted girls, but some also targeted other participants. Aside from the 16 cash and non-cash transfers, which almost always went to the household head (the household head may have been the adolescent girl themselves, but this was rarely clearly stated), half the interventions (n = 20) focused only on girls, and half targeted boys and girls (n = 20). Fifteen interventions targeted parents, spouses or the wider community of the adolescent girls, for example, with adult–youth and adult groups in DISHA [43]. 

### 3.4. Evaluations

Twenty interventions were evaluated using randomised controlled trials (RCTs), fourteen were non-randomised and eight were natural experiments using survey data (two studies used different designs in different areas).

There was variation in the timing of endline outcome data collection, from immediately after the intervention ended, e.g., Ishraq Pilot [40], to eight years later, e.g., the Ghanaian School Scholarship Programme [39].

For the majority of interventions (n = 30), pregnancy or birth were used as outcome measures. Twenty studies measured contraceptive use and nine included other related measures, such as ideal number of children or unmet need for family planning. 

## 4. Discussion

A range of structural interventions aiming to address upstream factors have been evaluated in terms of their impact on adolescent contraceptive use and/or pregnancy/birth. Furthermore, aside from the variation in the intervention content, there is diversity in the populations targeted and settings. There is also diversity of evaluations, in terms of the study design, follow-up period and outcome measures. This heterogeneity makes synthesis or reaching a consensus about “what works” difficult. This creates challenges for policy makers and practitioners—it can be hard to judge which intervention activities would be the most feasible and effective in their specific context.

The interventions’ mechanisms of action were often unclear; for example, cash transfers could work by reducing poverty, by incentivising certain behaviours and/or by elevating the status of the person it was conditional for (i.e., the adolescent girl in this instance). Vocational training could reduce poverty by leading to employment or income-generating activities, but it could also increase autonomy, raise aspirations, reduce social isolation and build self-confidence. A better understanding of how interventions work will enable greater learning from outcome evaluations—not just to explore which activities should be incorporated, but how best they could be adapted to suit a new context. Future evaluations should explicitly test interventions’ mechanisms of action, so that we are able to judge not just whether to replicate an intervention, but how to scale it up or introduce it into a new context. Since replication of such interventions can rarely be completely faithful to the original, either in design, implementation or the effect it has in a new context, it is crucial that we understand what are the key mechanisms through which it has an effect. This will allow attention to be placed on ascertaining whether these mechanisms have been replicated, even if the intervention activities, population or setting, are different from the original evaluation. Intervention evaluations should incorporate process evaluations for this purpose, as well as to capture implementation and contextual information that could further help to understand why or how an intervention was (or was not) effective. The subsequent phase of our project aims to explore these issues, in order to develop a mid-range theory that could be operationalised in a variety of settings and with different adolescent sub-populations.

Other systematic reviews have either included both structural and non-structural interventions (e.g., [9,44,45]) or have included a broader range of outcomes than just contraception/childbearing (e.g., [17,46,47]). Other reviews have also noted the range of outcome measures and study designs used in evaluations of structural or adolescent contraception/childbearing interventions [44,47]. This map extends the evidence gap map conducted by 3ie, not only by updating it, but also by looking more in-depth at structural contraceptive interventions specifically [17].

A limitation of this map stems from the lack of consensus around what constitutes a structural intervention, as well as challenges around classifying interventions as structural or not, based on sometimes limited information in the available documentation. As such, we may have excluded interventions that others consider structural, or included some that others would not consider structural. A further limitation was that the search was limited to English and Portuguese articles. Although we did not identify any Portuguese papers, we may have missed articles in other languages, or grey literature from Portuguese or other non-English web pages. Nevertheless, we are not aware of any other review that has identified the number and range of structural interventions evaluating contraceptive/childbearing outcomes as we have. This supports our belief that a strength of our systematic approach to identifying studies is its comprehensiveness and its inclusion of grey literature from a number of sources. Others have noted the importance of this, particularly for structural interventions [45]. Finally, by omitting abortion as an outcome, we may have missed pertinent studies (however, even if it were included, data would be under-reported since abortion is illegal in many of the included countries).

## 5. Conclusions

A range of structural interventions have been evaluated for their effect on adolescent contraceptive use and pregnancy. These interventions, and their evaluations, are heterogenous in numerous ways. A better understanding of how different structural interventions work, as well as addressing the challenges of evaluating interventions, including which outcome measures are most appropriate, is needed to facilitate progress in enabling adolescent contraceptive use in LMICs.

## Figures and Tables

**Table 1 ijerph-19-11715-t001:** Exclusion criteria.

Exclusion Criteria	Description of Criteria
Year Published	Exclude if published before 2005.
Country	Exclude if the intervention was NOT conducted in low- and middle-income countries, as defined by the World Bank in 2019.
Topic	Exclude if not about sexual or reproductive health.
Study design	Exclude if not an intervention evaluation.
Outcomes	Exclude if not reporting at least one of the following outcomes: - Uptake or use of modern contraception (evaluations reporting condom use only were only included if the intervention clearly stated a goal of pregnancy prevention and condoms were used for contraceptive purposes or for dual protection); - Intention/readiness to use contraception; - Desire to avoid, delay, space or limit childbearing; - Desire to use contraception; - Pregnancy/birth.
Participants	Exclude if not focused on adolescents aged 10–19 years (only include if the intervention either targeted 10–19-year-olds, or at least 50% of study sample were aged 10–19 years, or the mean or median age was 19 years or younger, or results were presented separately for this age group).
Intervention focus	Exclude if intervention does not focus on structural interventions (girls’ economic or other empowerment, school enrolment and retention, shaping norms around gender, sexual behaviour or fertility, advocacy and other interventions to reduce gender and other inequalities).

**Table 2 ijerph-19-11715-t002:** Different types of structural interventions evaluated.

Type of Structural Intervention	N
Poverty reduction/economic empowerment	29
Encouraging school participation	17
Changing community social norms	13

**Table 3 ijerph-19-11715-t003:** Types of poverty reduction/economic empowerment activities evaluated.

Poverty Reduction/Economic Empowerment Activity	N
Financial literacy training	14
Vocational or livelihoods training	12
Conditional cash transfer	12
Savings accounts	9
Microfinance	6
Unconditional cash transfer	5
Non-cash transfer	5
Employment or income-generating opportunities	3

## Data Availability

Not applicable.

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
