# Peer review of "Which Structural Interventions for Adolescent Contraceptive Use Have Been Evaluated in Low- and Middle-Income Countries?"

_ijerph, 2022, doi:10.3390/ijerph191811715_

Round 1

Reviewer 1 Report

This is a wonderful manuscript. I really enjoyed reading the content.  It is an interesting and important study that highlighting the importance of structural interventions for adolescent contraceptive use. 

I offer a few minor suggestions that may help your manuscript:

·      How the authors define structural interventions?     Maybe they need to incorporate a definition in 3.2 

·      One limitation more is that the search was limited to English or Portuguese languages for low and middle-incomes countries and maybe there are others interventions in the local languages. 

Author Response

Reviewer 1

Response

This is a wonderful manuscript. I really enjoyed reading the content.  It is an interesting and important study that highlighting the importance of structural interventions for adolescent contraceptive use. 

Thank you for your positive feedback!

How the authors define structural interventions?     Maybe they need to incorporate a definition in 3.2 

We have added a definition on p2: “Structural interventions target the structural-level factors i.e. “the physical, social, cultural, organizational, community, economic, legal, or policy aspects of the environment” (p1) that can affect health and contraceptive behaviours{Sumartojo, 2000 #143} . “

One limitation more is that the search was limited to English or Portuguese languages for low and middle-incomes countries and maybe there are others interventions in the local languages.

We agree that this is a relevant point and have added it to our discussion of the study’s limitations on page 7: “A further limitation was that the search was limited to English and Portuguese articles. Although we did not identify any Portuguese papers, we may have missed articles in other languages, or grey literature from Portuguese or other non-English web pages.”

Reviewer 2 Report

Thank you for submitting your manuscript:  Which structural interventions for adolescent contraceptive use 2 have been evaluated in low- and middle-income countries? Sometimes this research are difficult to see why they are important but this work is. If Sustainable 35 Development Goal 3, “to ensure healthy lives and promote well-being for all at all ages” is to see change bigger structural issues must be accountable and research findings meaningful, not just the same work for example gender equality and government spending. This work made me think, thank you.

It took me a a second read to see how meaningful this work is. (also, I may have been distracted).  I also wondered why abortion as an outcome was not included. This work might be used to develop future programs and not keep repeating interventions that are not relevant to the real world and lived experience of girls and young women. Perhaps you have future research ready to go !?

Author Response

Reviewer 2

Response

Thank you for submitting your manuscript:  Which structural interventions for adolescent contraceptive use 2 have been evaluated in low- and middle-income countries? Sometimes this research are difficult to see why they are important but this work is. If Sustainable 35 Development Goal 3, “to ensure healthy lives and promote well-being for all at all ages” is to see change bigger structural issues must be accountable and research findings meaningful, not just the same work for example gender equality and government spending. This work made me think, thank you.

It took me a a second read to see how meaningful this work is. (also, I may have been distracted).  I also wondered why abortion as an outcome was not included.

This work might be used to develop future programs and not keep repeating interventions that are not relevant to the real world and lived experience of girls and young women. Perhaps you have future research ready to go !?

Thank you for your positive comments!

We did not include abortion as an outcome since the project was originally funded to look at contraception (indeed it was for this reason that our subsequent, in depth analysis, focused only on contraceptive outcomes). However we agree that you make a good point, as we included pregnancy and birth outcomes, so abortion outcomes would also be relevant. That said, even if we did include abortion, data would typically be under-reported as it would be illegal in many of the included countries. We have mentioned this in our discussion of the study’s limitations on page 7: “Finally, by omitting abortion as an outcome, we may have missed pertinent studies (however even if it were included, data would be under-reported since abortion is illegal in many of the included countries).”

In terms of future research, we are about to submit an in-depth analysis of the contraceptive studies included in the map for publication in this journal.

Reviewer 3 Report

IJERPH Review

 Which structural interventions for adolescent contraceptive use have been evaluated in low- and middle-income countries?

This manuscript proposes a comprehensive, well-presented review of existing evaluations of interventions targeting adolescents living in LMIC and aiming to improve contraceptive / unwanted-pregnancy indicators.

Overall, the paper is clearly written and the evaluation selection and review methodology is detailed enough to be easily replicable. Some additional analysis and edits may help further strengthen the value of this paper before it is published.

·       The authors brush upon the lack of clarity in the “interventions’ mechanisms of action” (l. 217 – 218) but it might be worth getting into more aspects of these limitations and how they affect this study design: structural factors can almost systematically affect the primary outcomes included here (i.e. contraceptive use, pregnancies, etc) in multiple ways and their effects will often be mediated by other variables (e.g. contraceptive supply environment, those same individual level factors that are evaluated elsewhere) through feedback loops, positive and negative externalities and interfering / confounding variabls es. Overall, isolating the exact effect of any given structural intervention on adolescents’ contraceptive use may be nearly impossible (which is why the findings of this manuscript are inconclusive)?

·       The authors list several upstream factors that could affect contraceptive use among adolescents, but reviews only seem to include interventions targeting gender norms, economic empowerment, and schooling. What about the others (e.g. fertility norms, poverty reduction)

·       p. 80 et al. Did the authors try to assess the “robustness” of their findings by modifying the inclusion and exclusion criteria for the participants (i.e. either making them more stringent or more flexible)?

·       p. 103 Could the geographic distribution of the reviewed evaluation be skewed by the language inclusion criteria (only English and Portuguese articles were reviewed)? This should appear as a limitation in the following section.

·       We would have expected to see the section on the gender distribution of intervention targets a bit higher in the results section.

·       Overall, the results are extremely descriptive. It might have been interesting to cross-tabulate some of the variables (and if at all possible, look at associations) such as intervention types and evaluation designs or primary outcomes.

·       In the discussion section, the authors suggest that “future evaluations should explicitly test intervention mechanism of action”. It may also be important to encourage process evaluations in order to capture implementation vs design failures and dose-response aspects of these interventions.

Author Response

Reviewer 3

Response

This manuscript proposes a comprehensive, well-presented review of existing evaluations of interventions targeting adolescents living in LMIC and aiming to improve contraceptive / unwanted-pregnancy indicators.

Overall, the paper is clearly written and the evaluation selection and review methodology is detailed enough to be easily replicable. Some additional analysis and edits may help further strengthen the value of this paper before it is published.

Thank you for your positive comments!

The authors brush upon the lack of clarity in the “interventions’ mechanisms of action” (l. 217 – 218) but it might be worth getting into more aspects of these limitations and how they affect this study design: structural factors can almost systematically affect the primary outcomes included here (i.e. contraceptive use, pregnancies, etc) in multiple ways and their effects will often be mediated by other variables (e.g. contraceptive supply environment, those same individual level factors that are evaluated elsewhere) through feedback loops, positive and negative externalities and interfering / confounding variabls es. Overall, isolating the exact effect of any given structural intervention on adolescents’ contraceptive use may be nearly impossible (which is why the findings of this manuscript are inconclusive)?

We get into more detail about mechanisms of actions in our linked, in-depth analysis paper (which we are soon to submit for publication).

The authors list several upstream factors that could affect contraceptive use among adolescents, but reviews only seem to include interventions targeting gender norms, economic empowerment, and schooling. What about the others (e.g. fertility norms, poverty reduction)

This is a good point – we have revised our terminology to be more clear as I believe we referred to gender norms when we meant social norms (i.e. gender or fertility, or potentially norms around adolescent sexuality). Similarly, some of the economic empowerment interventions may have been focused on poverty reduction rather than empowerment per se (we had taken the view that poverty reduction in itself would be empowering, but can see that this isn’t necessarily the case). We have revised this too.

p. 80 et al. Did the authors try to assess the “robustness” of their findings by modifying the inclusion and exclusion criteria for the participants (i.e. either making them more stringent or more flexible)?

I’m not sure what this point is referring to, since we didn’t have participants (nor a page 80). In terms of our exclusion criteria, we were aware that if we’d made them more flexible, we could have included many sex education intervention evaluations, however we did not consider these to be structural. Similarly, there were many studies that did not focus on, or provide results specifically for, adolescents, but captured data on a wider age range.

 p. 103 Could the geographic distribution of the reviewed evaluation be skewed by the language inclusion criteria (only English and Portuguese articles were reviewed)? This should appear as a limitation in the following section.

We agree – and have added this on p7: “A further limitation was that the search was limited to English and Portuguese articles. Although we did not identify any Portuguese papers, we may have missed articles in other languages, or grey literature from Portuguese or other non-English web pages.”

We would have expected to see the section on the gender distribution of intervention targets a bit higher in the results section.

We understand your perspective – however to move it earlier would mean presenting this prior to the findings about what intervention had been evaluated. We think it makes sense to present aims and content first.

Overall, the results are extremely descriptive. It might have been interesting to cross-tabulate some of the variables (and if at all possible, look at associations) such as intervention types and evaluation designs or primary outcomes.

We understand your concern – however as the aim of this paper was to map structural interventions, by its nature it will be descriptive.

In the discussion section, the authors suggest that “future evaluations should explicitly test intervention mechanism of action”. It may also be important to encourage process evaluations in order to capture implementation vs design failures and dose-response aspects of these interventions.

This is a really good point – we have added this in to page 6: “. Intervention evaluations should incorporate process evaluations for this purpose, as well as to capture implementation and contextual information that could further help to understand why or how an intervention was (or was not) effective. “